# Coordinate-Aware Modulation for Neural Fields

**Joo Chan Lee**[1]**, Daniel Rho**[2]**, Seungtae Nam**[1]**, Jong Hwan Ko**[1✉]**, Eunbyung Park**[1✉]
[1]Sungkyunkwan University, [2]KT
{maincold2,stnamjef,jhko,epark}@skku.edu, daniel.r@kt.com

## Abstract

Neural fields, mapping low-dimensional input coordinates to corresponding signals, have shown promising results in representing various signals. Numerous methodologies have been proposed, and techniques employing MLPs and grid representations have achieved substantial success. MLPs allow compact and high expressibility, yet often suffer from spectral bias and slow convergence speed. On the other hand, methods using grids are free from spectral bias and achieve fast training speed, however, at the expense of high spatial complexity. In this work, we propose a novel way for exploiting both MLPs and grid representations in neural fields. Unlike the prevalent methods that combine them sequentially (extract features from the grids first and feed them to the MLP), we inject spectral bias-free grid representations into the intermediate features in the MLP. More specifically, we suggest a Coordinate-Aware Modulation (CAM), which modulates the intermediate features using scale and shift parameters extracted from the grid representations. This can maintain the strengths of MLPs while mitigating any remaining potential biases, facilitating the rapid learning of high-frequency components. In addition, we empirically found that the feature normalizations, which have not been successful in neural filed literature, proved to be effective when applied in conjunction with the proposed CAM. Experimental results demonstrate that CAM enhances the performance of neural representation and improves learning stability across a range of signals. Especially in the novel view synthesis task, we achieved state-of-the-art performance with the least number of parameters and fast training speed for dynamic scenes and the best performance under 1MB memory for static scenes. CAM also outperforms the best-performing video compression methods using neural fields by a large margin. Our project page is available at https://maincold2.github.io/cam/.

## 1 Introduction

Neural fields (also known as coordinate-based or implicit neural representations) have attracted great attention (Xie et al., 2022) in representing various types of signals, such as image (Chen et al., 2021b; Mehta et al., 2021), video (Rho et al., 2022; Chen et al., 2022b), 3D shape (Tancik et al., 2020; Chabra et al., 2020), and novel view synthesis (Mildenhall et al., 2020; Barron et al., 2021; 2022). These methods typically use a multi-layer perceptron (MLP), mapping low-dimensional inputs (coordinates) to output quantities, as shown in Fig. 1-(a). It has achieved a very compact representation by representing signals with the dense connections of weights and biases in the MLP architecture. However, a notable drawback of MLPs is their inherent spectral bias (Rahaman et al., 2019), which leads them to learn towards lower-frequency or smoother patterns, often missing the finer and high-frequency details. Despite the recent progress, such as frequency-based activation functions (Sitzmann et al., 2020) and positional encoding (Tancik et al., 2020), deeper MLP structures and extensive training duration are needed to achieve desirable performances for high-frequency signals (Mildenhall et al., 2020).

With fast training and inference time, the conventional grid-based representations (Fig. 1-(b)) have been recently repopularized in neural fields literature. They can represent high-frequency signals effectively (w/o MLPs or w/ small MLPs, hence no architectural bias), achieving promising reconstruction quality (Fridovich-Keil et al., 2022; Chan et al., 2022; Takikawa et al., 2022). However,

---

✉ Corresponding authors.

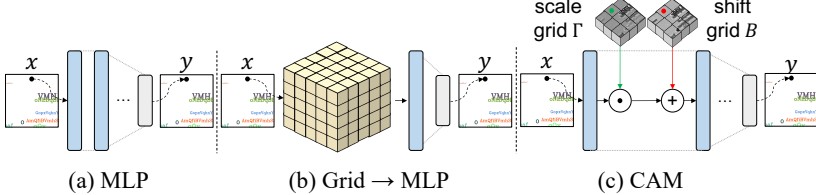

Figure 1: Feature representations based on the (a) MLP, (b) Grid → MLP, (c) CAM. The dot in CAM means a Hadamard product.

the grid structures (typically representing volume features with high resolution and large channels) cause a dramatic increase in memory footprints. Although many recent works have explored reducing the memory usage through grid factorization (Chen et al., 2022a; Fridovich-Keil et al., 2023), hash encoding (Müller et al., 2022), or vector quantization (Takikawa et al., 2022), constructing compact yet powerful grid representation remains a challenge.

A typical approach of leveraging both grids and MLPs is to combine them sequentially (Müller et al., 2022; Yu et al., 2021a; Takikawa et al., 2022), extracting the feature from the grid representations first and feeding them to MLPs. MLPs in these approaches play a secondary role in representing signals, and the small-size MLPs are generally used to finalize or refine the features from the grids. Therefore, the grids represent most of the signals' contents, and higher resolutions of the grids are required to achieve better performance, resulting in significant memory requirements.

In this work, we propose a novel way of exploiting grid representations in neural fields. Based on MLP architectures, we suggest a coordinate-aware modulation (CAM), which modulates intermediate features of the neural networks using the grids (Fig. 1-(c)). More specifically, CAM extracts scale and shift parameters from the grid representations given the input coordinates, then multiplies the extracted scale parameters to the intermediate features in MLPs and adds the shift parameters. Since CAM utilizes an interpolation scheme commonly used in recent grid representations, it can extract scale and shift parameters at any arbitrary location. The main idea behind the proposed CAM is to inject *spectral bias-free* representations into the intermediate features in MLPs. It will assist in mitigating any remaining potential biases in MLPs and help them quickly learn high-frequency components.

In addition, we found that feature normalization techniques (Ioffe & Szegedy, 2015; Ulyanov et al., 2016) proved to be effective when applied in conjunction with the proposed CAM. Normalizing intermediate features in neural fields has yet to show meaningful gains in the representation performance. However, without normalization techniques, training deep neural networks in general often requires careful learning rate schedules and other hyperparameter searches (Bjorck et al., 2018), and we observed similar phenomena in training neural fields. Given the same network architecture and task (training Mip-NeRF), different learning rate schedules resulted in significant performance variations (further discussed in App. C.2). We have demonstrated that CAM benefits from the feature normalizations, showing fast and stable convergence with superior performance.

We have extensively tested the proposed method on various tasks. The experimental results show that CAM improves the performance and robustness in training neural fields. First, we demonstrate the effectiveness of CAM in simple image fitting and generalization tasks, where CAM improved the baseline neural fields by a safe margin. Second, we tested CAM on video representation, applying CAM to one of the best-performing frame-wise video representation methods, and the resulting method set a new state-of-the-art compression performance among the methods using neural fields and frame-wise representations. We also tested CAM on novel view synthesis tasks. For static scenes, CAM has achieved state-of-the-art performance on real scenes (360 dataset) and also showed the best performance under a 1MB memory budget on synthetic scenes (NeRF synthetic dataset). Finally, we also tested on dynamic scenes, and CAM outperformed the existing methods with the least number of parameters and fast training speed (D-NeRF dataset).

## 2 RELATED WORKS

**Neural fields**, or implicit neural representations, use neural networks to represent signals based on coordinates. Recent studies on neural fields have shown promising results in a variety of vision

tasks such as image representation (Sitzmann et al., 2020; Dupont et al., 2021), video representation (Rho et al., 2022; Chen et al., 2021a), 3D shape representation (Tancik et al., 2020; Chabra et al., 2020; Park et al., 2019; Mescheder et al., 2019; Martel et al., 2021), novel view synthesis (Mildenhall et al., 2020; Barron et al., 2021; Müller et al., 2022; Fridovich-Keil et al., 2022; Yu et al., 2021a; Chen et al., 2022a; Yu et al., 2021b), and novel view image generation (Schwarz et al., 2020; Chan et al., 2021; Gu et al., 2022; Deng et al., 2022). Neural networks (typically using MLPs in neural fields) tend to be learned towards low-frequency signals due to the spectral bias (Rahaman et al., 2019). Several studies have been conducted to mitigate this issue by proposing frequency encodings (Mildenhall et al., 2020; Tancik et al., 2020; Barron et al., 2021) or periodic activations (Sitzmann et al., 2020; Mehta et al., 2021). Nevertheless, this challenge persists in the literature, demanding the use of complex MLPs and extensive training time to effectively represent high-frequency signals (Mildenhall et al., 2020).

An emerging alternative to this MLP-dependent paradigm is the use of an auxiliary data structure, typically grids, incorporated with interpolation techniques. Such approach has notably reduced training times without sacrificing the reconstruction quality (Fridovich-Keil et al., 2022; Chan et al., 2022; Takikawa et al., 2022). However, these grid frameworks, usually designed with high-resolution volumetric features, demand extensive memory consumption as shown in Fig. 1-(b). While numerous studies have made efforts to minimize memory usage via grid factorization (Chen et al., 2022a; Fridovich-Keil et al., 2023), pruning (Fridovich-Keil et al., 2022; Rho et al., 2023), hashing (Müller et al., 2022), or vector quantization (Takikawa et al., 2022), the pursuit of memory-efficient grid representation remains an ongoing focus in the field of neural fields research.

**Combination of an MLP and grid representation.** The aforementioned grid-based methods generally use a small MLP to obtain the final output from the grid feature. In other words, the grid structure and an MLP are sequentially deployed. Most recently, NFFB (Wu et al., 2023) proposed combining two architectures in a different way, by designing each of multiple sets of MLPs and grids to represent different frequency signals, similar to the concept of wavelets. Nonetheless, it is worth noting that NFFB demands task-specific designs for individual models. In contrast, CAM is a plug-and-play solution that can be easily deployed without the need for any modifications to the original model configurations.

**Modulation in neural fields.** Feature modulation in neural networks has been a well-established concept, spanning across diverse domains including visual reasoning (Perez et al., 2018), image generation (Ghiasi et al., 2017; Chen et al., 2019), denoising (Mohan et al., 2021), and restoration (He et al., 2019). They typically employ an additional network (or linear transform) to represent modulation parameters, learning a well-conditional impact on the intermediate features of the base network. Neural fields literature follows the paradigm by representing modulation parameters with the function of noise vector (Pi-GAN (Chan et al., 2021)), datapoints (COIN++ (Dupont et al., 2022)), patch-wise latent feature (ModSiren (Mehta et al., 2021)), or input coordinate (MFN, FINN (Fathony et al., 2021; Zhuang, 2024)). In contrast to other methods that integrate periodic functions into their approach, both FINN and our proposed method utilize coordinate-dependent parameters to directly influence the intermediate features. However, while FINN acts as a filter by using the same vector for all layers, our model represents different scale and shift (scalar) values in each layer. Furthermore, the utilization of grid representation for scale and shift parameters in our model avoids introducing any network architectural bias. This is distinct from all the aforementioned methods, which can induce architectural bias by incorporating a separate linear layer following positional encoding.

## 3 METHOD

**Coordinate-aware modulation.** Existing literature has delved into the representation of signals presenting continuous yet diverse characteristics across arbitrary coordinates, mainly relying on either neural networks (typically MLPs) or grid structures. We rethink this paradigm and propose coordinate-aware modulation (CAM), which combines both architectures in parallel. CAM exploits the implicit representation of neural networks, which ensures compactness regardless of high-dimensional coordinates, while representing high-frequency signals effectively using grids. More specifically, CAM modulates the intermediate features of the neural networks based on input coordinates, where the scalar factors for scale and shift modulation are parameterized by grids. CAM can retain compactness since the grids represent single-channel modulation parameters, different from

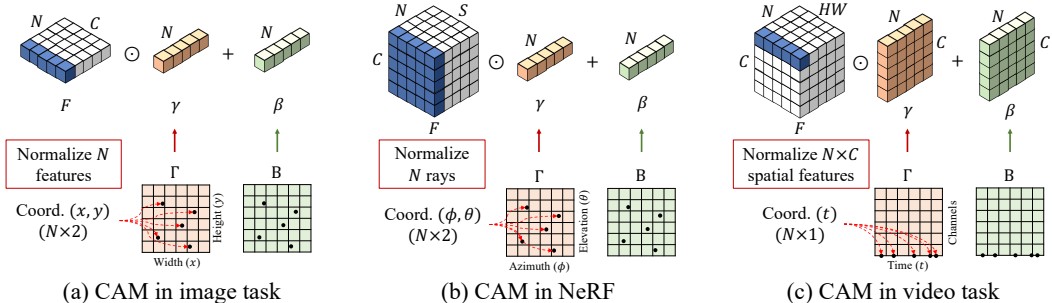

Figure 2: Visualization of CAM on different domains.

the general use of grids that represent large-channel features with high resolution. Formally, taking a 1D feature of an MLP as an example, the formulation is as follows,

$$\tilde{F}_{n,c} = \gamma_n(X; \Gamma)F_{n,c} + \beta_n(X; B), \tag{1}$$

where $F, \tilde{F} \in \mathbb{R}^{N \times C}$ are an intermediate feature tensor and the modulated output feature ($N$: batch size, $C$: channel size), and $n, c$ denote the batch and channel index of the feature, respectively. $\gamma(\cdot; \Gamma), \beta(\cdot; B) : \mathbb{R}^{N \times D} \to \mathbb{R}^N$ are the scale and shift function of input coordinates $X \in \mathbb{R}^{N \times D}$ ($D$: input coordinate dimension), outputting scalar values from the single-channel grids $\Gamma, B$ given each coordinate. $\gamma_n(\cdot; \Gamma), \beta_n(\cdot; B)$ denote each scale and shift factor for batch $n$.

**Coordinate priority for CAM.** The grids are adopted to parameterize single-channel features (modulation parameters), but they can face challenges with the curse of dimensionality, especially with high-dimensional input coordinates (e.g., 6-dimensional coordinates for dynamic NeRFs). To avoid the complex dimension of grids, we strategically prioritize which coordinates to use for representing modulation parameters into grids, among diverse coordinates of each task. Visual signals can have several dimensions, including space, viewing direction, and time. At the core, spatial components construct distinct scenes, where the viewing directions determine which aspect of the scene becomes visible, and the temporal coordinates represent dynamic movements in the scene. Among the view direction and time coordinates, we empirically found that considering temporal coordinates is more beneficial for CAM. This can be interpreted that a visible scene determined by spatiality and view direction, is the basis of effectively defining a time-varying scene. We establish this hierarchy of coordinates, prioritizing the highest-level components among the coordinates (denoted as $X^{(\cdot)}$) to be regarded for modulation (e.g., temporal coordinates $X^{(t)}$ for dynamic NeRFs and view direction coordinates $X^{(\phi,\theta)}$ for NeRFs). Given that image and frame-wise video representations involve only spatial and time coordinates, respectively, we use the complete input coordinate by denoting it as $X$, in the following sections.

**Feature normalization.** We standardize the intermediate feature $F$ with its mean and variance before applying the modulation. Although general neural representation methods cannot take advantage of feature normalization due to its regularizing property for fitting, we empirically found that normalization integrated with CAM facilitates and stabilizes model convergence. We hypothesize that the enforcing diverse distribution of standardized features acts as de-regularization, which stands for fitting signals. We compute the mean and variance along with as many dimensions as possible, excluding the batch dimension.

Although CAM serves as a universal method that can be applied to any neural fields for a wide variety of tasks, each task possesses its unique characteristics and intermediate feature shapes. In the following sections, we will provide a more in-depth explanation of the CAM approach for each specific task.

## 3.1 IMAGE

We can formulate a neural field as a function of a 2-dimensional coordinate that outputs the corresponding color in order to represent images (Sitzmann et al., 2020; Tancik et al., 2020). When a stack of 2-dimensional coordinates $X \in \mathbb{R}^{N \times 2}$ pass through a neural network, CAM normalizes and modulates a latent feature $F^l \in \mathbb{R}^{N \times C}$ of each layer $l$, where $C$ is the channel (or feature) size (we will omit superscript $l$ for brevity). As images only have spatial coordinates, we obtain

the modulation parameters corresponding to these coordinates. More precisely, Fig. 2-(a) illustrates how CAM works in the task, and CAM can be formally written as follows:

$$\tilde{F}_{n,c} = \gamma_n(X;\Gamma)\frac{F_{n,c} - \mu_n(F)}{\sqrt{\sigma_n^2(F) + \epsilon}} + \beta_n(X;B),$$
(2)

$$\mu_n(F) = \frac{1}{C}\sum_c F_{n,c}, \quad \sigma_n^2(F) = \frac{1}{C}\sum_c (F_{n,c} - \mu_n(F))^2,$$
(3)

where $F, \tilde{F} \in \mathbb{R}^{N \times C}$ are latent and modulated latent feature tensors, respectively. The mean and variance functions $\mu(\cdot), \sigma^2(\cdot) : \mathbb{R}^{N \times C} \to \mathbb{R}^N$ normalize features over every dimension except for the batch dimension. Similarly, the scale and shift functions $\gamma(\cdot;\Gamma), \beta(\cdot;B) : \mathbb{R}^{N \times 2} \to \mathbb{R}^N$ output scalar values for each coordinate, and $\gamma_n(\cdot;\Gamma), \beta_n(\cdot;B)$ denote each scale and shift factor for batch $n$. We can extract values from the grid representations for scale and shift parameters ($\Gamma, B \in \mathbb{R}^{d_x \times d_y}$, $d_x$ and $d_y$ are the grid resolutions) by bilinearly interpolating values using neighboring input coordinates $X$.

## 3.2 NOVEL VIEW SYNTHESIS

**Neural radiance fields (NeRFs).** A NeRF model uses an MLP architecture to model a function of a volume coordinate $(x, y, z)$ and a view direction $(\phi, \theta)$ that outputs RGB color $c$ and density $d$. To calculate the color of each pixel (camera ray), a NeRF samples $S$ points along the ray and aggregates color and density values of the sampled points using the volume rendering equation (Mildenhall et al., 2020). Since outputs of sampled points in a ray will be merged to get the color of a ray, we view a pack of points per ray as a single unit. It constructs an input coordinate tensor $X \in \mathbb{R}^{N \times S \times 5}$, and latent features $F \in \mathbb{R}^{N \times S \times C}$. Based on the proposed priority, CAM is applied for NeRFs according to the view directional coordinates of $N$ ray units $X^{(\phi,\theta)} \in \mathbb{R}^{N \times 2}$ (Fig. 2-(b)), formally defined as follows:

$$\tilde{F}_{n,s,c} = \gamma_n(X^{(\phi,\theta)};\Gamma)\frac{F_{n,s,c} - \mu_n(F)}{\sqrt{\sigma_n^2(F) + \epsilon}} + \beta_n(X^{(\phi,\theta)};B),$$
(4)

$$\mu_n(F) = \frac{1}{SC}\sum_{s,c} F_{n,s,c}, \quad \sigma_n^2(F) = \frac{1}{SC}\sum_{s,c}(F_{n,s,c} - \mu_n(F))^2,$$
(5)

where $\mu(\cdot), \sigma^2(\cdot) : \mathbb{R}^{N \times S \times C} \to \mathbb{R}^N$ denote mean and variance functions, and $\mu_n(F)$ and $\sigma_n^2(F)$ represent the mean and variance for ray $n$ when $F$ is given. As mentioned in Sec. 3, we normalize over all dimensions except for the batch size. $\gamma(\cdot;\Gamma), \beta(\cdot;B) : \mathbb{R}^{N \times 2} \to \mathbb{R}^N$ are scale and shift functions, parameterized by two grid representations $\Gamma, B \in \mathbb{R}^{d_\phi \times d_\theta}$; $d_\phi$ and $d_\theta$ are resolutions of azimuth $\phi$ and elevation $\theta$ dimension, respectively. Similar to $\mu_n(F)$ and $\sigma_n^2(F)$, the scalars $\gamma_n(X;\Gamma)$ and $B_n(X;B)$ denote the scale and shift value, respectively, for ray $n$.

**Dynamic NeRFs** build upon the static NeRFs concept by introducing the ability to model time-varying or dynamic scenes, representing 4D scenes that change over time (Pumarola et al., 2021). This is achieved by adding a time coordinate $t$ to the input of the NeRFs. Therefore, the overall process for CAM follows as in Eq. 4, except that the modulation parameters are obtained corresponding to time coordinates $X^{(t)} \in \mathbb{R}^{N \times 1}$, from two 1-dimensional grids $\Gamma, B \in \mathbb{R}^{d_t}$ ($d_t$ is the resolution of the temporal dimension).

## 3.3 VIDEO

Videos can be represented as a function of temporal and spatial coordinates. However, this pixel-wise neural representation demands significant computational resources and time, limiting its practical use (Chen et al., 2021a). To tackle the challenges associated with high computational costs and slow training/inference times, NeRV (Chen et al., 2021a) and its variations (Li et al., 2022; Lee et al., 2023) adopted a frame-wise representation approach and use neural fields as a function of only the temporal coordinate $t$. This not only accelerated training and inference time but also improved compression and representation performance (Chen et al., 2021a). These frame-wise video representation models leverage convolutional layers to generate a video frame per temporal coordinate $t$. More precisely, an input coordinate tensor $X \in \mathbb{R}^{N \times 1}$ associated with $N$ temporal coordinates is

Table 1: Performance evaluation for image regression and generalization measured in PSNR.

| Method | #Params | Regression | | Generalization | |
|---|---|---|---|---|---|
| | | *Natural* | *Text* | *Natural* | *Text* |
| I-NGP | 237K | **32.98** | 41.94 | 26.11 | 32.37 |
| FFN | 263K | 30.30 | 34.44 | 27.48 | 30.04 |
| + CAM | 266K | 32.21 (+1.91) | **50.17** (+15.73) | **28.19** (+0.71) | **33.09** (+3.05) |

Table 2: Effectiveness in the NeRF task. * denotes the reported value in the original paper.

| Method | #Params | Time | PSNR |
|---|---|---|---|
| NerfAcc | 0.6M | 38 m | 31.55 |
| K-planes | 37M | 38* m | 32.36 |
| CAM | 3.7M | 51 m | 32.18 |
| | 13M | 54 m | 32.60 |

supplied to the neural network to generate intermediate feature tensors $F \in \mathbb{R}^{N \times C \times H \times W}$, where $N, C, H$ and $W$ denote the number of frames or batch size, the number of channels, the feature's height and width, respectively. Then, we can define CAM as follows:

$$\tilde{F}_{n,c,h,w} = \gamma_{n,c}(X; \Gamma) \frac{F_{n,c,h,w} - \mu_{n,c}(F)}{\sqrt{\sigma_{n,c}^2(F) + \epsilon}} + \beta_{n,c}(X; B), \qquad (6)$$

$$\mu_{n,c}(F) = \frac{1}{HW} \sum_{h,w} F_{n,c,h,w}, \quad \sigma_{n,c}^2(F) = \frac{1}{HW} \sum_{h,w} (F_{n,c,h,w} - \mu_{n,c}(F))^2, \qquad (7)$$

where $\mu(\cdot)$, $\sigma^2(\cdot) : \mathbb{R}^{N \times C \times H \times W} \to \mathbb{R}^{N \times C}$ denote mean and variance functions. The reason for not normalizing over every dimension except the batch dimension is to keep the computational costs affordable (see App. B). Motivated by Ulyanov et al. (2016), we exclude the channel dimension, and represent channel-wise modulation parameters by scale and shift functions $\gamma(\cdot; \Gamma), \beta(\cdot; B)$ : $\mathbb{R}^{N \times 1} \to \mathbb{R}^{N \times C}$. The grids for scales and shifts are denoted by $\Gamma$ and $B$, where $\Gamma$ and $B$ are of size $\mathbb{R}^{d_t \times C}$, respectively. Here, $d_t$ represents the grid resolution in the time dimension, and $C$ represents the channel size which is the same as the channel size in the feature tensor $F$. Fig. 2-(c) illustrates how CAM works in frame-wise video representation neural fields.

## 4 EXPERIMENTS

We initially assessed the effectiveness of CAM in terms of mitigating spectral bias. Then, we evaluated our proposed method on various signal representation tasks, including image, video, 3D scene, and 3D video representations. Finally, we delved into the reasons behind its superior performance, conducting comprehensive analyses. All baseline models were implemented under their original configurations, and CAM was applied in a plug-and-play manner. CAM includes feature normalization throughout the experiments, except for efficient NeRFs (e.g., NerfAcc (Li et al., 2023)), where we found that the normalization is ineffective for pre-sampled inputs. We provide implementation details for each task in App. A.

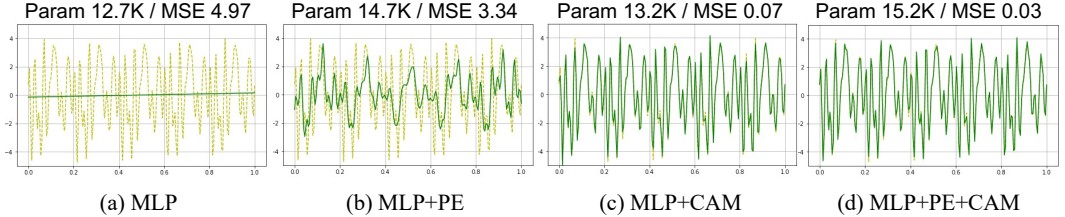

| Param 12.7K / MSE 4.97 | Param 14.7K / MSE 3.34 | Param 13.2K / MSE 0.07 | Param 15.2K / MSE 0.03 |
|---|---|---|---|
| (a) MLP | (b) MLP+PE | (c) MLP+CAM | (d) MLP+PE+CAM |

Figure 3: Performance on 1D signal regression. The yellow dotted line represents GT.

### 4.1 MOTIVATING EXAMPLE

We begin by demonstrating the spectral bias-free representation of CAM evaluated on 1D sinusoidal function regression (Fig. 3). The figure indicates that the MLP is not capable of representing the high-frequency signal, even though positional encoding (PE) is applied. In contrast, when CAM is applied to the MLP, the resulting model successfully represents the signal, even with less parameter overheads compared to PE. The model applying both PE and CAM shows the most accurate

representation. These results demonstrate that CAM can be an effective solution for resolving the spectral bias of the MLP while maintaining compactness.

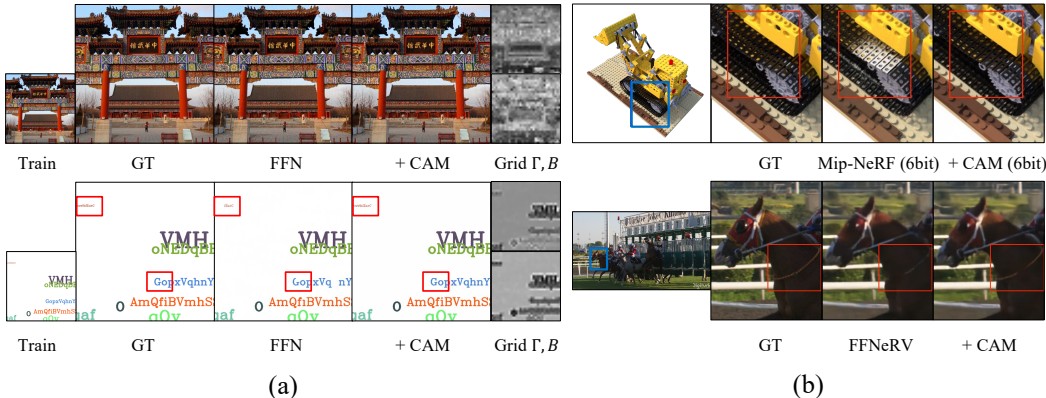

Figure 4: Qualitative results on (a) image generalization task with visualization of grids extracted from the last hidden layer and (b) novel view synthesis and video representations.

## 4.2 RESULTS ON VARIOUS TASKS

**Image.** Tab. 1 shows the results of two subtasks, image regression and generalization. CAM improves performance of FFN on both tasks across two image datasets, with negligible additional parameters. While I-NGP demonstrated impressive results in *Natural* image regression, highlighting its superiority in overfitting, it fell short in terms of image generalization. In contrast, CAM consistently shows high performance in both tasks, demonstrating its overall effectiveness. Fig. 4-(a) shows the quality results for image generalization. As the figure shows, both scale and shift grids ($\Gamma$, $B$) reflect the shape of the entire image, indicating the grids effectively represent the signal. Especially for the text image, CAM allows distinguishing the text and background, resulting in significantly increased performance.

Table 3: Qualitative results evaluated on NeRFs. The sizes are measured in megabytes (MB).

| Bit | Method | NeRF Synthetic | | NSVF Synthetic | | LLFF | |
|---|---|---|---|---|---|---|---|
| | | Size | PSNR | Size | PSNR | Size | PSNR |
| 32 | NeRF | 5.00 | 31.01 | 5.00 | 30.81 | 5.00 | 26.50 |
| | TensoRF | 71.9 | 33.14 | ≈ 70 | 36.52 | 179.7 | 26.73 |
| | Mip-NeRF | **2.34** | 33.09 | **2.34** | 35.83 | **2.34** | 26.86 |
| | + CAM | **2.34** | **33.42** | **2.34** | **36.56** | **2.34** | **27.17** |
| 8 | Rho et al. | 1.69 | 32.24 | 1.88 | 35.11 | 7.49 | 26.64 |
| | TensoRF | 16.9 | 32.78 | 17.8 | 36.11 | 44.7 | 26.66 |
| | Mip-NeRF | **0.58** | 32.86 | **0.58** | 35.52 | **0.58** | 26.64 |
| | + CAM | **0.58** | **33.27** | **0.58** | **36.30** | **0.58** | **26.88** |

Table 4: Performance evaluation on the 360 dataset, which comprises unbounded real scenes. Among 9 scenes, we evaluate 7 publicly available scenes. CAM is applied on Mip-NeRF 360.

| Method | #Params | PSNR |
|---|---|---|
| Mip-NeRF | **0.6M** | 25.12 |
| I-NGP | 84M | 27.06 |
| Zip-NeRF | 84M | 29.82 |
| Mip-NeRF 360 | 9M | 29.11 |
| + CAM | 9M | **29.98** |

**Novel view synthesis on static scene.** We first present the superiority of CAM over representations based on an MLP or grid with a small MLP using the NeRF synthetic dataset. As the baseline models, we adopted NerfAcc (Li et al., 2023) and K-planes (Fridovich-Keil et al., 2023) for MLP- and grid-based representations (Fig.1-(a),(b)), respectively. We modulate the intermediate features of NerfAcc, utilizing modulation parameters represented by tri-plane factorized grids with a singular channel. For a fair comparison with K-planes, here we refrained from implementing our proposed priority and used spatial coordinates to represent modulation parameters. As shown in Tab. 2, CAM outperforms other baselines, resulting in the best visual quality with compactness and comparable training duration, validating its efficiency.

We also evaluated with more powerful baseline models, Mip-NeRF and Mip-NeRF 360. Tab. 3, 4 show the qualitative results for the NeRF synthetic, NSVF, LLFF, and real 360 datasets. Throughout all the datasets, CAM showcases significant improvement in PSNR, with a negligible increase

in the number of parameters. Especially for the 360 dataset, CAM achieves state-of-the-art performance. We also tested on lower bit precision; we quantized every weight parameter including $\Gamma$ and $B$. As Tab. 3 shows, CAM exhibits robustness to lower bit precision and remains effective. Furthermore, the CAM-applied 8-bit model consistently outperforms the 32-bit original Mip-NeRF. Consequently, CAM achieves state-of-the-art performance under a 1MB memory budget on NeRF synthetic dataset, as shown in Fig. 5. As the qualitative results using *Lego* (Fig. 4-(b)) shows, the baseline performs poor reconstruction containing an incorrectly illuminated area while the CAM-applied model reconstructs accurately. This indicates that modulation according to view directions results in robustness for representing view-dependent components.

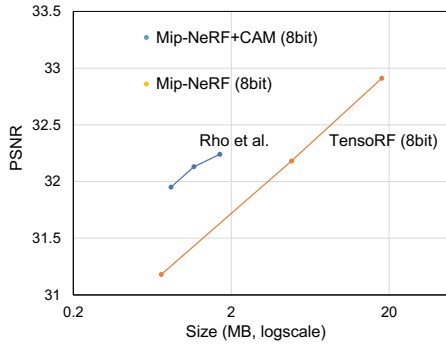

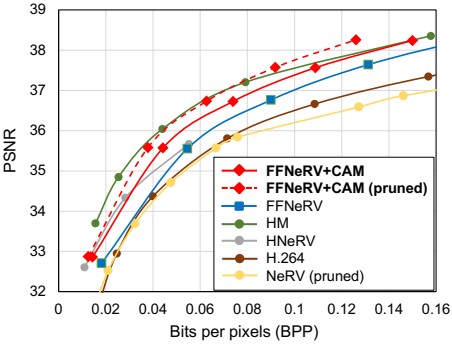

Figure 5: The rate-distortion curve evaluated on NeRF synthetic dataset.

Figure 6: The rate-distortion curve on UVG dataset (best viewed in color).

**Dynamic scene.** We used the D-NeRF dataset (Pumarola et al., 2021) to evaluate CAM for novel view synthesis under dynamic scenes, as shown in Tab. 5. CAM is applied on NerfAcc (Li et al., 2023) for T-NeRF (a variant of D-NeRF). CAM sets a new benchmark, outperforming the previous state-of-the-art by more than 1 PSNR, even while using the least parameters. Furthermore, our model is time-efficient, needing only an hour for training, thanks to its foundation on Nerfacc that boasts rapid processing due to efficient sampling.

Table 5: Performance evaluation of dynamic NeRFs.

| Method | #Params | PSNR |
|---|---|---|
| D-NeRF | 1.1M | 29.67 |
| TiNeuVox | 12M | 32.67 |
| K-planes | 37M | 31.61 |
| NerfAcc | **0.6M** | 32.22 |
| + CAM | **0.6M** | **33.78** |

**Video.** In Fig. 4-(b), the qualitative results for video representation highlight the enhanced visual quality achieved by CAM. We offer detailed results of video representation performance in App. C.1, and here, we focus on showcasing video compression performance, a central and practical task for videos. Fig. 6 visualize the rate-distortion for video compression. In the range from low to high BPP, CAM improves compression performance compared to the baseline FFNeRV by a significant margin. It achieves comparable performance with HM, the reference software of HEVC (Sullivan et al., 2012). Distinct from HEVC, a commercial codec designed under the consideration of time efficiency, HM shows significantly high performance under heavy computations. HM has a decoding rate of around 10 fps using a CPU (Hu et al., 2023), while our model is built on FFNeRV (Lee et al., 2023), a neural representation capable of fast decoding, allowing for real-time processing with a GPU (around 45 fps at 0.1 BPP). To our knowledge, our compression performance is state-of-the-art among methods that have the capability for real-time decoding.

## 4.3 ANALYSIS AND ABLATION STUDIES ON CAM

**Motivation.** We analyzed the intermediate feature distribution in the image generalization task, where the features can be visualized straightforwardly, as depicted in Fig. 7-(a). CAM shows a high variance of pixel-wise features while improving the visual quality. This observation underscores the idea that the representation power can be boosted when the features of different coordinates become more distinct from each other. CAM is a strategic approach to achieve this, while it maintains compactness by representing only modulating scalar factors into grids.

**Mitigating spectral bias.** We visualized the error map in the frequency domain (Fig. 7-(b)) to validate that CAM is actually capable of representing high-frequency components. CAM reduces the errors in high frequency noticeably with only negligible grid parameters (263K for the MLP vs. 3K for the grid in Tab. 1), indicating its effective mitigation of the MLP's spectral bias.

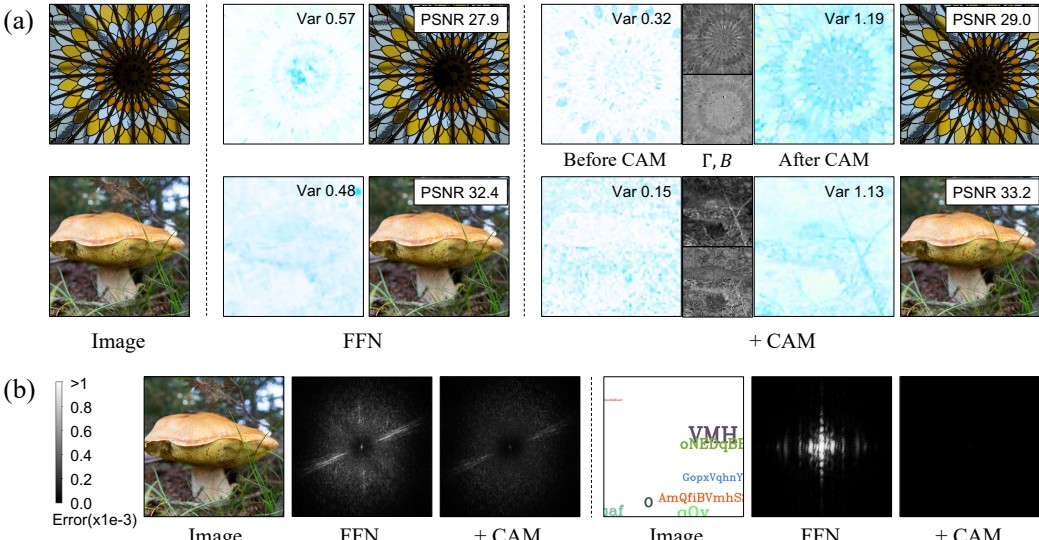

Figure 7: (a) Visualization of the pixel-wise distribution before and after applying CAM on the final feature. The same color indicates the same distribution (mean and variance). We provide variance between pixels of each feature (described in App. A) and output PSNR. (b) Error map in the frequency domain: A more centralized pixel of the maps indicates an error in the lower frequency.

**Coordinate priority.** As shown in Tab. 6, CAM with the highest-level coordinates based on the proposed priority achieves the optimal performance. CAM with spatial coordinates is effective for modalities with only spatial coordinates (images), as we have shown in Tab. 1. However, when the input modality becomes more complex in NeRFs and dynamic NeRFs, spatiality-aware modulation can be meaningless in spite of the requirement of large additional memory (even with the factorized grids). Furthermore, although using both time and view direction coordinates increases performance compared to the baseline in D-NeRF, a single prioritized component demonstrates the most efficient result.

Table 6: Ablation study on the proposed priority. S, D, and T denote space, direction, and time coordinates.

| Baseline | CAM coord. | | | #Params | PSNR |
|---|---|---|---|---|---|
| | S | D | T | | |
| NerfAcc (D-NeRF) | ✓ | | | 0.6M | 32.22 |
| | | | | 13.1M | 32.57 |
| | | ✓ | | 0.6M | 32.44 |
| | | ✓ | ✓ | 0.6M | 32.49 |
| | | | ✓ | **0.6M** | **33.78** |
| Mip-NeRF | ✓ | | - | 0.6M | 33.09 |
| | | | - | 13.1M | 32.70 |
| | | ✓ | - | **0.6M** | **33.42** |

**Effect of feature normalization.** As shown in Tab. 7, normalization with CAM consistently enhances the performance for diverse tasks, while naively applying normalization typically degrades performance. In addition, CAM allows one of the known advantages of normalization, decreasing the magnitude of gradients and improving convergence speed (Ioffe & Szegedy, 2015), further discussed in App. C.2.

Table 7: Ablation study on the feature normalization, evaluated on *Natural* images, *Ready* video, and *Lego* scene. CAM-N indicates CAM without normalization.

| Task | Base | BN | LN | IN | CAM-N | CAM |
|---|---|---|---|---|---|---|
| Image | 30.3 | 23.6 | 30.8 | - | 30.9 | **32.2** |
| Video | 31.6 | 22.1 | - | 31.5 | 31.9 | **32.3** |
| NeRF | 35.7 | 35.2 | 35.4 | - | 35.9 | **36.2** |

## 5 CONCLUSION

We have proposed a Coordinate-Aware Modulation (CAM), a novel combination of neural networks and grid representations for neural fields. CAM modulates the intermediate features of neural networks with scale and shift parameters, which are represented in the grids. This can exploit the strengths of the MLP while mitigating any architectural biases, resulting in the effective learning of high-frequency components. In addition, we empirically found that the feature normalizations, previously unsuccessful in neural field literature, are notably effective when integrated with CAM. Extensive experiments have demonstrated that CAM improves the performance of neural representations and enhances learning stability across a wide range of data modalities and tasks. We renew state-of-the-art performance across various tasks while maintaining compactness. We believe it opens up new opportunities for designing and developing neural fields in many other valuable applications.

ACKNOWLEDGEMENTS

This work was supported by the Ministry of Science and ICT (MSIT) of Korea under the National Research Foundation (NRF) grants (RS-2023-00245342) and Institute of Information and Communication Technology Planning Evaluation (IITP) grants (IITP-2019-0-00421, IITP-2023-2020-0-01821, IITP-2021-0-02068), and by the Technology Innovation Program (RS-2023-00235718) funded by the Ministry of Trade, Industry & Energy (1415187474).

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

APPENDIX

# A IMPLEMENTATION DETAILS

In this section, we provide a brief explanation of the functional form of grids and specify our implementation details for diverse tasks.

## A.1 FUNCTION OF $\Gamma$ AND $B$

$\Gamma$ and $B$ are grid structures to represent the scale and shift factors, where each grid is trained as a function of coordinates with infinite resolution, outputting coordinate-corresponding components. The output in infinite resolution is aggregated by nearby features in the grid based on the distance between the coordinates of the input and neighboring features.

## A.2 1D SIGNAL

We conducted the experiment for regressing 1D periodic function, following the previous works (Rahaman et al., 2019; Cho et al., 2022). We constructed the target function $f(x) = \sum_{i=1}^{10} \sin(2\pi k_i x + \phi_i)$, where $k_i \in \{5, 10, ..., 50\}$, $\phi_i \sim U(0, 2\pi)$ and we uniformly sampled $x$ in the range of $[0, 1]$. The learning rate was set to $10^{-3}$ and we trained for 1500 iterations using the Adam optimizer. We used a 4-layer MLP with 64 channels as the baseline and also set the grid resolution of 64. When applying PE, we enlarged the single-channel coordinate to 32 channels, and we concatenated the original and enlarged inputs.

## A.3 IMAGE

For the 2D image representation task, we used *Natural* and *Text* image datasets (Tancik et al., 2020), which include $512 \times 512$ images, respectively. The resolution of the grids ($d_x$ and $d_y$) was set to $32 \times 32$. Using two subtasks, we assessed the ability to regress the training data points and to generalize well on unseen data points. The first subtask is to accurately represent a target image at a resolution of $512 \times 512$, using the same image for training, and it aims to measure the ability for fitting signals. Another subtask trains neural fields using a smaller image with a resolution of $256 \times 256$, but evaluates using the original image with a resolution of $512 \times 512$.

We used FFN (Tancik et al., 2020) as the baseline model, which was originally developed in Jax a few years back. Due to its older environment, we opted for a Pytorch implementation to simplify the experimental process. We constructed a baseline model following the original paper (MLP with 4 layers, 256 hidden channels, ReLU activation, and sigmoid output). Each model was trained for 2000 iterations using the Adam optimizer. The learning rate was initially set to $10^{-3}$ and $10^{-2}$ for neural networks and grids, respectively, multiplied by 0.1 at 1000 and 1500 iterations. The manually tuned parameters for each dataset in FFN were also used in this experiment, where the gaussian scale factor was set to 10 and 14 for *Natural* and *Text*, respectively.

For I-NGP (Müller et al., 2022), we used hash grids with 2-channel features across 16 different resolutions (16 to 256) and a following 2-layer 64-channel MLP. The maximum hash map size was set to $2^{15}$.

**The variance in Fig. 7(a)** denotes the mean of the variance of all pixels at the same channel. Formally, the variance $v$ of $H \times W$ pixel-wise $C$-channel features $X \in \mathbb{R}^{C \times H \times W}$ can be expressed as, $v = \frac{1}{C} \sum_{ch=1}^{C} var^{(H,W)}(X_{ch})$, where $var^{(H,W)}(\cdot) : \mathbb{R}^{H \times W} \to \mathbb{R}$ computes the variance of $H \times W$ values and $X_{ch} \in \mathbb{R}^{H \times W}$ is features at the channel $ch$.

## A.4 NOVEL VIEW SYNTHESIS

**Static scene.** We used synthetic (NeRF (Mildenhall et al., 2020), NSVF (Liu et al., 2020)), forward-facing (LLFF (Mildenhall et al., 2019)), and real-world unbounded (360 (Barron et al., 2022)) datasets for evaluating novel view synthesis performance. As a baseline model, we used Mip-NeRF (Barron et al., 2021) for single-scale scenes, except for 360 dataset, where we used Mip-NeRF 360 (Barron et al., 2022). We implemented CAM based on Mip-NeRF and Mip-NeRF 360 official

Table 8: Compression performance evaluated on UVG videos at various levels. BPP denotes "bits per pixel".

| Video (#frames) | Beauty (600) | Bospho (600) | Honey (600) | Ready (600) | Jockey (600) | Shake (300) | Yacht (600) | Avg. |
|---|---|---|---|---|---|---|---|---|
| PSNR | 33.65 | 34.59 | 38.89 | 33.89 | 27.1 | 33.43 | 28.76 | 32.86 |
| BPP | 0.0144 | 0.0149 | 0.0148 | 0.0142 | 0.0145 | 0.0115 | 0.0148 | 0.0144 |
| PSNR | 34.21 | 38.39 | 39.58 | 37.29 | 31.62 | 35.18 | 32.53 | 35.57 |
| BPP | 0.0454 | 0.0459 | 0.0442 | 0.0439 | 0.0448 | 0.0355 | 0.0455 | 0.0442 |
| PSNR | 34.51 | 39.87 | 39.71 | 38.32 | 33.64 | 36.65 | 34.35 | 36.73 |
| BPP | 0.0752 | 0.0751 | 0.0728 | 0.0721 | 0.0735 | 0.0743 | 0.0748 | 0.0739 |
| PSNR | 34.78 | 40.91 | 39.86 | 38.92 | 35.23 | 37.24 | 35.84 | 37.56 |
| BPP | 0.1122 | 0.1109 | 0.1087 | 0.1068 | 0.1089 | 0.0980 | 0.1108 | 0.1088 |
| PSNR | 35.06 | 41.71 | 40.01 | 39.3 | 36.5 | 37.71 | 37.13 | 38.24 |
| BPP | 0.1563 | 0.1530 | 0.15114 | 0.1480 | 0.1508 | 0.1249 | 0.1535 | 0.1500 |

codes in the Jax framework. While following all the original configurations, we incorporated CAM into every MLP linear layer until the view direction coordinates were directly inputted. For the scale and shift grids ($\Gamma$, $B$), the values of $d_\theta$ and $d_\phi$ were set to 4 and 3 for forward-facing scenes, and 10 and 3 for other scenes, respectively. For quantization, we applied layer-wise min-max quantization-aware training (QAT), as in Rho et al. (2023). We compared our method with NeRF (Mildenhall et al., 2020), TensoRF (Chen et al., 2022a), Rho et al. (2023) for NeRF synthetic, NSVF, and LLFF datasets in Tab. 3, and with I-NGP (Müller et al., 2022) and Zip-NeRF (Barron et al., 2023) for 360 dataset in Tab 4.

**Dynamic scene.** We used the D-NeRF dataset (Pumarola et al., 2021) to evaluate CAM for novel view synthesis under dynamic scenes. CAM was implemented on NerfAcc (Li et al., 2023) with the grid resolution $d_t$ of 10. NerfAcc for dynamic scene was originally based on T-NeRF (Pumarola et al., 2021), which deploys deformation network and canonical network. We incorporated CAM into every linear layer in the canonical network, until the view direction coordinates were directly inputted. We compared our approach with the baseline NerfAcc and recent state-of-the-art algorithms for dynamic NeRF (D-NeRF (Pumarola et al., 2021), TiNeuVox (Fang et al., 2022), and K-planes. (Fridovich-Keil et al., 2023)).

## A.5 VIDEO

**Video representation.** To measure the video representation performance of neural fields, we used the UVG dataset (Mercat et al., 2020), which is one of the most popular datasets in neural field-based video representation. The UVG dataset contains seven videos with a resolution of 1920 × 1080. Among video representing neural fields (Chen et al., 2021a; Li et al., 2022; Lee et al., 2023), we used FFNeRV (Lee et al., 2023) as our baseline model because of its compactness and representation performance. We implemented CAM based on FFNeRV official codes in the Pytorch framework. To ensure consistency, we maintained all the original configurations including QAT, with the exception of applying CAM between the convolutional and activation layers of each FFNeRV convolution block. In regards to the scale and shift grids ($\Gamma$, $B$), we set $d_T$ to 60 for both the 32-bit and 8-bit models, and 30 for the 6-bit model.

**Compression comparison.** For video compression results, we followed the compression pipeline used in FFNeRV, which includes QAT, optional weight pruning, and entropy coding. Although FFNeRV quantized to 8-bit width for model compression, we further lowered the bit width to 6-bit, except for the last head layer. This was done because CAM exhibits robust performance even with 6-bit, where the baseline FFNeRV shows poor performance, as shown in Tab. 10. Fig. 6 of the main paper depicts the rate-distortion performance of our approach compared with widely-used video codecs (H.264(Wiegand et al., 2003), HM (HEVC (Sullivan et al., 2012) test model)), neural video representations (FFNeRV (Lee et al., 2023), NeRV (Chen et al., 2021a), HNeRV (Chen et al.,

2023)). Detailed compression performances of our model without pruning for each UVG video at various levels are reported in Tab. 8.

Table 9: Performance evaluation under different settings for frame-wise video representation.

| Norm Unit | $\Gamma, B$ shape | PSNR | Params (M) | Time/Epoch (sec) |
|---|---|---|---|---|
| $(H, W)$ | $\mathbb{R}^{d_t \times C}$ | 32.25 | 11.4 | 69.1 |
| $(H, W)$ | $\mathbb{R}^{d_t}$ | 31.93 | 11.3 | 68.8 |
| $(C, H, W)$ | $\mathbb{R}^{d_t \times C}$ | 32.37 | 11.4 | 102.8 |
| $(C, H, W)$ | $\mathbb{R}^{d_t}$ | 32.39 | 11.3 | 104.0 |

## B    ADAPTATION FOR 4D TENSOR

We generally proposed to compute the mean and variance along with as many dimensions as possible excluding the batch dimension, and represent scalar features in grids $\Gamma, B$. However, we introduce some adaptations for 4D intermediate tensors in frame-wise video representation: excluding also the channel dimension and representing channel-wise modulation factors in the grids. This is because of heavy computation from the large normalization unit, which causes a dramatic increase in training time (about 50%), as shown in Tab. 9. When we exclude channel axis, representing channel-wise modulation factors shows better than representing scalar factors. It is worth noting that our general proposal achieves the best performance, highlighting the flexibility of CAM where we can trade performance and complexity.

## C    ADDITIONAL EXPERIMENTAL RESULTS

Table 10: PSNR on video representation. The leftmost column denotes the bit precision of neural networks. BPP denotes "bits per pixel".

| Bit | Method | Beauty | Bospho | Honey | Jockey | Ready | Shake | Yacht | Avg. | BPP |
|---|---|---|---|---|---|---|---|---|---|---|
| 32 | FFNeRV | 34.28 | 38.67 | 39.70 | 37.48 | 31.55 | 35.45 | 32.65 | 35.70 | 0.2870 |
| | + CAM | 34.29 (+0.01) | 38.86 (+0.19) | 39.69 (-0.01) | 37.82 (+0.34) | 32.25 (+0.70) | 35.47 (+0.02) | 33.03 (+0.38) | 35.95 (+0.25) | 0.2894 |
| 8 | FFNeRV | 34.21 | 38.41 | 39.60 | 37.29 | 31.48 | 35.26 | 32.48 | 35.55 | 0.0718 |
| | + CAM | 34.27 (+0.06) | 38.82 (+0.41) | 39.67 (+0.07) | 37.63 (+0.34) | 32.12 (+0.64) | 35.39 (+0.13) | 32.90 (+0.42) | 35.86 (+0.31) | 0.0723 |
| 6 | FFNeRV | 34.09 | 37.26 | 39.13 | 36.63 | 30.47 | 34.54 | 31.65 | 34.85 | 0.0538 |
| | + CAM | 34.21 (+0.12) | 38.25 (+0.99) | 39.21 (+0.08) | 37.16 (+0.53) | 31.57 (+1.10) | 35.02 (+0.48) | 32.50 (+0.85) | 35.45 (+0.60) | 0.0540 |

### C.1    VIDEO REPRESENTATION

Tab. 10 shows the video representation performance, measured in PSNR. The CAM-applied models consistently beat the baselines, regardless of videos. The performance gap is much wider for fast-moving videos (e.g., *Ready* and *Jockey*) than it is for static videos (e.g., *Beauty* and *Honey*). This result demonstrates that extended representational capacity in the temporal dimension due to grids surely improves performance in representing time-varying information. In addition, the performance gap between video representations with and without CAM widened as the bit precision decreased (from 0.25 to 0.60). These results imply that our method can be useful for neural fields designed for storage-constrained situations.

### C.2    EFFECT OF NORMALIZATION

In addition to the result in Tab. 7, we analyze the actual benefits of feature normalization in CAM. One of the known advantages of normalization is that it decreases the magnitude of gradients and

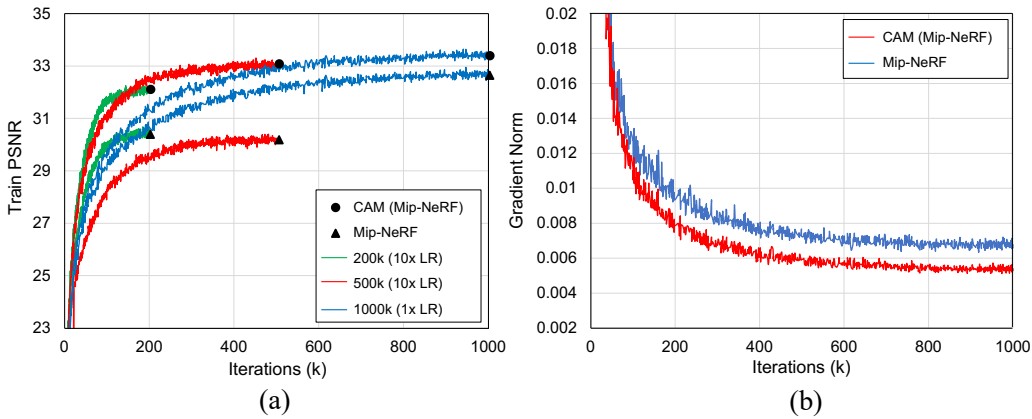

Figure 8: Analysis on convergence using *Lego* scene. (a) Train PSNRs with different learning schedules, while quantization-aware trained to 8-bit. (b) Gradient norm of weights during training.

prevents them from diverging, which allows the use of a higher learning rate and improved convergence speed (Ioffe & Szegedy, 2015). As shown by the decreased level of gradients in Fig. 8-(c), CAM benefits from the stabilizing advantage of normalization, achieving comparable and even superior performance to the baseline, with only 1/5 and half of the training duration, respectively (Fig. 8-(a)). Without CAM, different learning rate schedules resulted in significant performance variations (a learning rate schedule over 1,000K iterations vs. 500K iterations).

Table 11: Inference speed and GPU memory requirement of CAM compared to MLP-based and grid-based methods, using *Mic* scene.

| Method | Test chunk | PSNR | Inf. FPS | Inf. Mem. |
|---|---|---|---|---|
| K-Planes | - | 34.10 | 0.25 | 3.8 GB |
| Nerfacc
+CAM | 1024 | 33.77
36.03 | 0.51
0.26 | 4.4 GB
4.7 GB |
| Nerfacc
+CAM | 4096 | 33.77
36.03 | 1.19
0.67 | 10.5 GB
8.8 GB |
| Nerfacc
+CAM | 8192 | 33.77
36.03 | 1.45
1.01 | 19.5 GB
16.4 GB |

### C.3 INFERENCE SPEED AND MEMORY

We report the inference speed and GPU memory requirements of the models in Tab. 2, evaluated on the 'Mic' scene. As shown in Tab. 11, K-Planes requires small memory while showing slow inference. CAM reduces the original NerfAcc's speed when testing chunk size is small. However, increasing the testing chunk size reduces the speed gap between using CAM and not using it. Intriguingly, CAM even lowers memory usage under these conditions. We interpret that CAM facilitates a more effectively trained occupancy grid and helps bypass volume sampling, offsetting the additional computational demands introduced by CAM itself.

### C.4 PER-SCENE RESULTS.

We evaluated the performance on various datasets for novel view synthesis. We provide per-scene results for NeRF synthetic (Tab. 12), NSVF synthetic(Tab. 13), and LLFF (Tab. 14), 360 (Tab. 15), and D-NeRF (Tab. 16) datasets.

Table 12: Per-scene performance on the NeRF synthetic dataset measured in PSNR.

| Bit | Method | Chair | Drums | Ficus | Hotdog | Lego | Materials | Mic | Ship | Avg. |
|-----|--------|-------|-------|-------|--------|------|-----------|-----|------|------|
| 32 | Mip-NeRF | 35.14 | 25.48 | 33.29 | 37.48 | 35.70 | 30.71 | 36.51 | 30.41 | 33.09 |
| | + CAM | 35.24 | 25.74 | 34.07 | 37.89 | 36.24 | 31.48 | 36.04 | 30.64 | 33.42 |
| 8 | Mip-NeRF | 34.68 | 25.48 | 33.20 | 37.28 | 35.29 | 30.52 | 36.18 | 30.28 | 32.86 |
| | + CAM | 34.98 | 25.80 | 33.77 | 37.77 | 35.95 | 31.48 | 35.96 | 30.47 | 33.27 |

Table 13: Per-scene performance on the NSVF dataset measured in PSNR.

| Bit | Method | Bike | Lifestyle | Palace | Robot | Spaceship | Steamtrain | Toad | Wineholder | Avg. |
|-----|--------|------|-----------|--------|-------|-----------|------------|------|------------|------|
| 32 | Mip-NeRF | 38.51 | 34.77 | 37.00 | 36.65 | 38.09 | 36.94 | 33.58 | 31.12 | 35.83 |
| | + CAM | 39.06 | 35.21 | 37.41 | 37.70 | 41.24 | 37.49 | 33.59 | 30.77 | 36.56 |
| 8 | Mip-NeRF | 38.20 | 34.46 | 36.85 | 36.46 | 38.00 | 36.77 | 32.84 | 30.55 | 35.52 |
| | + CAM | 38.88 | 34.92 | 37.15 | 37.52 | 40.94 | 37.43 | 33.13 | 30.45 | 36.30 |

Table 14: Per-scene performance on the LLFF dataset measured in PSNR.

| Bit | Method | Fern | Flower | Fortress | Horns | Leaves | Orchids | Room | Trex | Avg. |
|-----|--------|------|--------|----------|-------|--------|---------|------|------|------|
| 32 | Mip-NeRF | 24.97 | 27.83 | 31.73 | 28.01 | 21.00 | 20.07 | 33.22 | 28.02 | 26.86 |
| | + CAM | 25.06 | 28.39 | 31.73 | 28.76 | 21.40 | 20.40 | 33.40 | 28.22 | 27.17 |
| 8 | Mip-NeRF | 24.95 | 27.56 | 31.27 | 27.66 | 20.88 | 20.07 | 32.97 | 27.73 | 26.64 |
| | + CAM | 25.06 | 27.72 | 31.45 | 28.18 | 21.27 | 20.37 | 33.13 | 27.88 | 26.88 |

Table 15: Per-scene performance on the 360 dataset measured in PSNR.

| Method | Bicycle | Bonsai | Counter | Garden | Kitchen | Room | Stump | Avg. |
|--------|---------|--------|---------|--------|---------|------|-------|------|
| Mip-NeRF 360 | 24.37 | 33.46 | 29.55 | 26.98 | 32.23 | 31.63 | 26.40 | 29.23 |
| + CAM | 24.30 | 35.44 | 30.62 | 26.99 | 33.60 | 32.91 | 26.03 | 29.98 |

Table 16: Per-scene performance on the D-NeRF dataset measured in PSNR.

| Method | Balls | Hell | Hook | Jacks | Lego | Mutant | Standup | Trex | Avg. |
|--------|-------|------|------|-------|------|--------|---------|------|------|
| NerfAcc | 39.49 | 25.58 | 31.86 | 32.73 | 24.32 | 35.55 | 35.90 | 32.33 | 32.22 |
| + CAM | 41.52 | 27.86 | 33.20 | 33.89 | 25.09 | 36.29 | 37.57 | 34.81 | 33.78 |

