# OpenReview forum: "Coordinate-Aware Modulation for Neural Fields"
_ICLR.cc/2024/Conference — ICLR 2024 spotlight_

### Official Review · Reviewer_DCsj · 2023-10-16

**Soundness:** 3 good
**Presentation:** 3 good
**Contribution:** 3 good
**Rating:** 8
**Confidence:** 4

**Summary:**

This main contribution of this paper is to use a grid-based approach to provide a scale and bias for the features generated at each layer of an implicit neural network, an approach used to encode any kind of signal. This is in contrast to the typical approach where the grid-based approach provides an input to the implicit neural network. In other words, the proposed approach is: a) use the input coordinate to recover the scale and bias for all layers of the MLP, b) recursively apply each layer of the network, normalize features, apply scale and bias for the appropriate layer. The authors also discuss which subset of input coordinates should be used to define the grid. Other contributions of the paper include the use of the aforementioned feature normalization in implicit neural representation and a comparison to various benchmarks on image encoding and generalization, novel view synthesis in static and dynamic NERF and video compression.

**Strengths:**

Originality:
* As far as I know, use a grid-based approach for scale/bias in implicit neural representation is new.

Quality:
* Evaluation is performed on several different tasks, with good results, thus I believe the strength of the method is demonstrated.
* The baselines used are generally competitive and recent.

Clarity:
* The paper is well written.

Significance:
* good results on multiple tasks, state-of-the-art in some.

**Weaknesses:**

Quality:
* I find the use of FFN (Tancik, 2020) as the only baseline in the image task disappointing. While not the most significant experiment of the paper, I think the use of more recent baselines and in particular of other grid-based approaches, for example at least instant-NGP, would make the comparison on images more significant. I also want to add that I find the baselines in the video experiment to be adequate, and arguably the video experiment is more important.

Clarity:
* I might have missed it, but for me, the paper does not sufficiently discuss/explain the reasoning behind the choice of coordinates that are used in the grid to select a scale and bias. For example, the scale/bias depends on the pixel coordinates only for a picture. As far as I understand, scale/bias are thus the same for each channel. But for the video, the scale/bias depend on pixel coordinates, time and channel. I do not understand why the channel becomes important for the video. I noticed the ablation study (Table 6), but it only covers the NERF experiments. This ablation study also does not discuss making the scale/bias depend on both direction and time.
* I do not understand what is the variance represented in Figure 8. Is it the variance between the elements of the input of the last layer of the MLP?
* I find the notation in equation 1 and subsequent equations a bit confusing. Both $\gamma_n$ and $\beta_n$ takes as input the full batch $X$. This suggests that any element of the vectors $\gamma$ and $\beta$ may depend on the full batch $X$. I suspect this is not the case due to how grid based approaches typically work but I am not sure. Would it make sense to change the notation to $\gamma(X_n;\Gamma)$ or  $\gamma(X^{(n)};\Gamma)$, to be closer to the notation already used in equation 4?

**Questions:**

* On page 8, the paper states that HM decodes at 10fps using a GPU. I was very surprised, because, as far as I know, HM does not use a GPU. I also could not find a mention of this fact in (Hu et al., 2023). While it is common to compare decoding speed on different hardware (CPU for traditional codec, GPU for ML methods), I think it is misleading to state that HM uses GPU. Could you please comment on/clarify this?
* Could you please comment on the choice of coordinates used in the grid to define scale/bias?
* Could you please provide further explanation about Figure 8?

---

> ### Author Response · Authors · 2023-11-20
> **(1/2) Thank you for all the invaluable comments and would like to respond individually.**
>
> We sincerely appreciate all insightful comments and efforts in reviewing our manuscript. All the reviewers acknowledged the innovative and impactful approach proposed in our paper, supported by extensive experiments. We intend to respond to each of your comments one by one, offering thorough explanations and clarifications as required.
>
> **[W1] More baseline (instant-NGP) for image tasks**
>
> Following the reviewer's suggestion, we assessed I-NGP for our image tasks. While I-NGP demonstrated impressive results in Natural image regression, highlighting its superiority in overfitting, it fell short in terms of image generalization. In contrast, CAM consistently shows high performance in both tasks, demonstrating its overall effectiveness. We will include this result in the upcoming draft.
>
> | Method | #Params | Natural (Reg.) | Text (Reg.) | Natural (Gen.) | Text (Gen.) |
> |:------:|---------|:--------------:|:-----------:|:--------------:|:-----------:|
> |  I-NGP | 237K    |      32.98     |    41.94    |      26.11     |    32.37    |
> | FFN    | 263K    |      30.30     |    34.44    |      27.48     |    30.04    |
> | + CAM  | 266K    |      32.21     |    50.17    |      28.19     |    33.09    |
>
> **[W2-1, Q2] Clarification of the choice for modulation parameters**
>
> CAM is applied not based on the signal's property but on the input coordinates. Although video signals contain spatial components, frame-wise methods such as FFNeRV only take time as the input coordinate to represent each corresponding video frame. This has been demonstrated as an efficient solution in terms of accuracy and training/inference speed, as opposed to incorporating both spatial and time coordinates (pixel-wise methods) in video representation literature. In our experiments, we adopted this more practical and applicable baseline and used the input time coordinate to determine the modulation parameters, showing a significant improvement over the baseline. Given that the time coordinate is the only option to determine the modulation parameters in this context, our ablation study presented in Table 6 did not include the video experiment. We would like to claim that this result highlights the wide and general application of CAM.
>
> **[W2-2] CAM based on both direction and time.**
>
> As described in Section 3.0, our approach involves assigning priority to input coordinates (when input coordinates include various components) for modulation parameters to avoid the curse of dimensionality. The prioritization is arranged in the following order: time, view direction, and space. This specific ordering is based on experimental results from tasks that involve several coordinate components, such as NeRF and dynamic NeRF, with the results detailed in Table 6. Furthermore, we have conducted additional evaluations of CAM regarding both direction and time coordinates in the following table. Although it increases performance compared to the baseline, using a single prioritized component demonstrates the most efficient result. This result will be included in the updated version.
> | S | D | T | #Params |  PSNR |
> |:-:|---|:-:|:-------:|:-----:|
> |   |   |   |   0.6M  | 32.22 |
> | V |   |   |  13.1M  | 32.57 |
> |   | V |   | 0.6M    | 32.44 |
> |   | V | V | 0.6M    | 32.49 |
> |   |   | V | 0.6M    | 33.78 |
>
> **[W3, Q3] Description of the variance in Figure 8**
>
> The variance in Figure 8 denotes the mean of the variance of all pixels at the same channel. Formally, the variance $v$ of $H\times W$ pixel-wise $C$-channel features $X \in \mathbb{R}^{C\times H\times W}$ can be expressed as follows,
>
> $v = {1\over C}\sum_{ch=1}^{C} var^{(H,W)}(X_{ch}),$
>
> where $var^{(H,W)}(\cdot):\mathbb{R}^{H\times W} \rightarrow \mathbb{R}$ computes the variance of $H\times W$ values and $X_{ch}\in\mathbb{R}^{H\times W}$ is features at the channel $ch$. We would like to clarify the information in the revised draft.

---

> ### Author Response · Authors · 2023-11-20
> **(2/2)**
>
> **[W4] Clarification of notation**
>
> In our paper, we have described the modulating parameter of MLP's intermediate features as $\gamma(X;\Gamma)$ (and $\beta(X;B)$), where the function $\gamma(\cdot;\Gamma):\mathbb{R}^{N\times D} \rightarrow \mathbb{R}^{N}$ and the input coordinate $X\in\mathbb{R}^{N\times D}$, with the dimension of the input coordinates $D$. However, we overlooked specifying $\gamma_n(X;\Gamma)$ as the scale value for batch $n$ in Equations 1 and 2, even though we precisely mention 'the scalars $\gamma_n(X; \Gamma)$ and $B_n(X; B)$ denote the scale and shift value, respectively, for ray $n$' in Equation 4.
>
> The notation $X^{(\cdot)}$ denotes the subset of input coordinates based on our proposed priority. For example, this could be the view-direction coordinate $X^{(\phi, \theta)}$ in NeRFs or the time coordinate $X^{(t)}$ in dynamic NeRFs. Given that image and video representations involve only spatial and time coordinates, respectively, we use the complete input coordinate by denoting it as $X$.
> We would like to revise and provide clarity on these aspects in the revised version of our draft.
>
> **[Q1] GPU usage of HM**
>
> We extend our sincere apologies for the incorrect statement that 'HM decodes at 10fps using a GPU'. Upon the reviewer's correct observation, we realize our misinterpretation of the report by Hu et al.[1] and acknowledge that HM natively supports only CPU usage, barring any specialized implementation. This error would be rectified in the revised version of our draft.
>
> [1] Hu, Z., Xu, D., Lu, G., Jiang, W., Wang, W., & Liu, S. (2022). FVC: An end-to-end framework towards deep video compression in feature space. IEEE Transactions on Pattern Analysis and Machine Intelligence, 45(4), 4569-4585.

---

> > ### Comment · Reviewer_DCsj · 2023-11-22
> > **Thank you for the update**
> >
> > Thank you for the update. I think my comments have all been addressed by the author. I think the paper looks better now. I will update my score.
> >
> > **W1**
> >
> > Thank you for the additional experiment. The results look good.
> >
> > **W2**
> > Thank you for the clarification and additional experiment. The results look quite nice and convincing.
> >
> > **W3**
> > Very clear, thank you.
> >
> > **W4**
> > Thank you for the explanation.
> >
> > **Q1**
> > Thank you for the clarification

---

### Official Review · Reviewer_7EJV · 2023-10-29

**Soundness:** 4 excellent
**Presentation:** 4 excellent
**Contribution:** 4 excellent
**Rating:** 8
**Confidence:** 1

**Summary:**

This work proposes combining grid input and MLP structure at intermediate features level for Neural fields application. This naturally extends MLP-only or grid-only methods, which despite its competitive performance have had downsides, such as not being able to represent high-frequency content or being computationally intensive.

**Strengths:**

- The biggest strength of CAM is its simplicity and its plug-and-play nature. I believe this will have much far-reaching impact in the Neural fields literature, compared to other highly sophisticated & implementation-heavy frameworks designed to maximize PSNR value at all costs. Similar to how widely Batch/layer-normalization has been used by the entire field.
- Extensive experimentation on diverse tasks and against different baselines add to the credibility of the work.

**Weaknesses:**

- While there are no specific weaknesses to point out, I don't think Figure 1 or Figure 2 convey the idea that well. Figure 1 probably will be better served by displaying more detailed mechanism (exammple of x, example of the values for \Gamma, etc.).
- Also giving a brief description of what functional form \Gamma and B take would be informative for readers.

**Questions:**

Don't have specific questions

---

> ### Author Response · Authors · 2023-11-20
> **Thank you for all the invaluable comments and would like to respond individually.**
>
> We sincerely appreciate all insightful comments and efforts in reviewing our manuscript. All the reviewers acknowledged the innovative and impactful approach proposed in our paper, supported by extensive experiments. We intend to respond to each of your comments one by one, offering thorough explanations and clarifications as required.
>
> **[W1] More informative figure**
> We appreciate the reviewer's thoughtful comment on our main figure. We revised Figure 1, as shown in [Link](https://raw.githubusercontent.com/anony3192/anony3192/main/figure1_rev.png). We would like to include this figure in the updated draft.
>
> **[W2] Brief description of the functional form $\Gamma$ and $B$**
>
> $\Gamma$ and $B$ are grid structures to represent the scale and shift factors, where each grid is trained as a function of coordinates with infinite resolution, outputting coordinate-corresponding components.
> The output in infinite resolution is aggregated by nearby features in the grid based on the distance between the coordinates of the input and neighboring features. We would like to include this in the appendix of the upcoming draft.

---

### Official Review · Reviewer_kCkz · 2023-11-01

**Soundness:** 3 good
**Presentation:** 3 good
**Contribution:** 3 good
**Rating:** 6
**Confidence:** 2

**Summary:**

The paper introduced a coordinate-aware modulation module that combines MLP features and grid representations for neural fields. Unlike the popular methods that chain the features, this new method not only preserves the strengths of MLP but also mitigates the bias problem by leveraging grid-based representation. The authors conducted experiments on tasks in multiple domains and the results demonstrate its capability of modeling high-frequency components and advantages over prevalent neural field features.

**Strengths:**

- The motivation of the paper was clearly stated
- The proposed approach is simple yet effective
- The paper is well structured and the idea is easy to follow
- Experiments are comprehensive. It covers various domains such as images, videos, etc.

**Weaknesses:**

- The numbers of the baseline models seem to be from the authors' own implementation, which makes it less appealing

**Questions:**

- Could you answer the first question I posted in the "Weaknesses" section?

---

> ### Author Response · Authors · 2023-11-20
> **Thank you for all the invaluable comments and would like to respond.**
>
> We sincerely appreciate all insightful comments and efforts in reviewing our manuscript. All the reviewers acknowledged the innovative and impactful approach proposed in our paper, supported by extensive experiments. We intend to respond to your comment, offering thorough explanations and clarifications as required.
>
> **[W1, Q1] Baseline model with our own implementation**
>
> For our experiments, we utilized the official codes for all baseline methods, including FFNeRV, NerfAcc, Mip-NeRF, and Mip-NeRF 360, and reported the values in the original papers. The only exception was FFN, which was originally developed in Jax a few years back. Due to its older environment, we opted for a Pytorch implementation to simplify the experimental process. We followed all the original settings of FFN, and successfully reproduced the reported value (1.9 PSNR higher for the Natural dataset and 0.4 lower for the Text dataset). It's important to note that, apart from the integration of CAM, all other configurations in our experiment were exactly the same. This consistency underscores the effectiveness of CAM in a clear and unbiased manner. If we understand your question correctly, we would like to include the reported values in the original paper and clarify this.

---

> > ### Comment · Reviewer_kCkz · 2023-12-04
> > **Thanks for the update**
> >
> > Thanks for your explanation! It might be a good idea to add a few words in the table caption to clarify you utilized the official code. It was a bit confusing without reference.

---

### Official Review · Reviewer_oV38 · 2023-11-02

**Soundness:** 3 good
**Presentation:** 4 excellent
**Contribution:** 3 good
**Rating:** 6
**Confidence:** 3

**Summary:**

Authors propose a new architecture for neural fields, i.e mapping low-dimensional input co-ordinates to the signal values, called CAM (co-ordinate aware modulation). The main idea is to modulate intermediate features using scale and shift parameters which are inferred from the low-dimensional input co-ordinates. Authors show that, while regular a regular MLP shows heavy spectral bias, and just grid representation is computationally very expensive, CAM can mitigate the spectral bias learning high frequency components, while also being compact. In addition, CAM facilitates adding normalization layers which improves training stability. Authors empirically show that CAM achieves competitive results image representation, novel view synthesis, and video representation tasks, while being fast and very stable to train.

**Strengths:**

+ Authors are tackling a very relevant problem, with wide interest to practitioners.
+ Paper is well written and easy to follow.
+ Claims in the paper are sounds. I particularly like that the argument about spectral bias and not learning high frequency components is verified empirically in Section 4.3
+ Experiments are sound and covers a wide range of tasks. Results are strong with performance comparable or exceeding the state of the art.

**Weaknesses:**

+ While the paper is generally strong, I believe that it lacks certain references which can put the work in a better context. There is a long history of using feature modulation in deep learning. A good example is [FiLM](https://arxiv.org/abs/1709.07871). This is also used for image generation/reconstruction tasks like in [generation](https://arxiv.org/abs/1810.01365), [denoising](https://arxiv.org/pdf/2107.12815.pdf), [image restoration](https://arxiv.org/abs/1904.08118) and [style transfer](https://arxiv.org/abs/1705.06830). Adding these references, and including a discussion around it can put feature modulation in a better context.

+ Can authors include inference speed/inference memory requirements to put regular MLP methods, grid based method, and CAM in prospective?

+ The choice of not using the all lower dimensional inputs to infer the scale and shift parameters, but a subset based on the problem is interesting. Have you conducted ablation studies that this is in some way beneficial?

A bit tangential but:
+ Do you think CAM can benefit decoder MLP for a triplane based representation as well? It would be cool to see some experiments and demonstrate the generality here.
+ In addition, I think text to 3D is one another domain where the speed and training stability of CAM can benefit quite a lot. If authors can demonstrate a couple of results comparing the training stability and speed using CAM augmented MLP in DreamFusion, that would be a great additon to the paper.

**Questions:**

See above.

---

> ### Author Response · Authors · 2023-11-20
> **Thank you for all the invaluable comments and would like to respond individually.**
>
> We sincerely appreciate all insightful comments and efforts in reviewing our manuscript. All the reviewers acknowledged the innovative and impactful approach proposed in our paper, supported by extensive experiments. We intend to respond to each of your comments one by one, offering thorough explanations and clarifications as required.
>
>
> **[W1] More references**
>
> We are grateful for the reviewer's insightful suggestion and would like to include the mentioned references for a more thorough discussion in our work.
>
>
>
> **[W2] Inference speed and memory**
>
> We report the inference speed and GPU memory requirements of the MLP-based method (NerfAcc), grid-based method (K-Planes), and CAM, evaluated on the 'Mic' scene. K-Planes requires small memory while showing slow inference. CAM reduces the original NerfAcc's speed when testing chunk size is small. However, increasing the testing chunk size reduces the speed gap between using CAM and not using it. Intriguingly, CAM even lowers memory usage under these conditions. We interpret that CAM facilitates a more effectively trained occupancy grid and helps bypass volume sampling, offsetting the additional computational demands introduced by CAM itself.
>
> |  Method  | Test chunk |  PSNR | Inf. FPS | Inf. Mem. |
> |:--------:|:----------:|:-----:|:--------:|:---------:|
> | K-Planes |      -     | 34.10 |   0.25   |   3.8 GB  |
> |  NerfAcc |    1024    | 33.77 |   0.51   |   4.4 GB  |
> |   +CAM   |    1024    | 36.03 |   0.26   |   4.7 GB  |
> |  NerfAcc |    4096    | 33.77 |   1.19   |  10.5 GB  |
> |   +CAM   |    4096    | 36.03 |   0.67   |   8.8 GB  |
> |  NerfAcc |    8192    | 33.77 |   1.45   |  19.5 GB  |
> |   +CAM   |    8192    | 36.03 |   1.01   |  16.4 GB  |
>
>
>
> **[W3] Ablation on the choice for modulation parameters**
>
> As described in Section 3.0, our approach involves assigning priority to input coordinates for modulation parameters to avoid the curse of dimensionality. The prioritization is arranged in the following order: time, view direction, and space. This specific ordering is based on experimental results from tasks that involve several coordinate components, such as NeRF and dynamic NeRF, with the results detailed in Table 6. Furthermore, we have conducted additional evaluations of CAM regarding both direction and time coordinates in the following table. Although it increases performance compared to the baseline, using a single prioritized component demonstrates the most efficient result.
> | S | D | T | #Params |  PSNR |
> |:-:|---|:-:|:-------:|:-----:|
> |   |   |   |   0.6M  | 32.22 |
> | V |   |   |  13.1M  | 32.57 |
> |   | V |   | 0.6M    | 32.44 |
> |   | V | V | 0.6M    | 32.49 |
> |   |   | V | 0.6M    | 33.78 |
>
>
>
> **[W4] CAM on tri-plane representation**
>
> When applied CAM to a tri-plane method (K-Planes), there is no significant improvement observed (from 34.10 to 34.12 in the 'Mic' scene). This is because general grid-based methods avoid spectral bias by utilizing grids with high resolution and large channels, where the decoder MLP primarily serves to aggregate these grid features. This is the reason for the substantial memory demands of grid-based methods. In contrast, CAM represents a novel approach to mitigate spectral bias, achieving this with notably compact parameters.
>
>
>
> **[W5] Application for DreamFusion**
>
> We have applied CAM to the official DreamFusion codebase, observing some impact on the outcome, as shown in Figure [Link](https://raw.githubusercontent.com/anony3192/anony3192/main/figure.png). However, due to time limitations, we were unable to conduct a more in-depth exploration. Exploring this area further, we believe, would constitute an interesting direction for future research.

---

### Author Response · Authors · 2023-11-22
**Update PDF**

Dear reviewers and AC,

We appreciate the reviewers' constructive comments on our manuscript. In response to the comments, we have carefully revised and enhanced the manuscript with the following aspects:

**Section 1**

(7EJV) More informative Figure 1

**Section 2**

(oV38) Long history of modulation in deep learning

**Section 3**

(DCsj) Clarification of notations

**Section 4**

(DCsj) More baseline for image tasks

(DCsj) Revising the statement for HM decoding

(oV38, DCsj) More ablation study for coordinate choice

**Appendix A**

(7EJV) Describing the function of grids

(kCkz) Clarification of our implementation for FFN

(DCsj) Description of the variance in Figure 8

**Appendix C**

(oV38) Evaluation of inference speed and memory
+ Due to the page limit, we relocated Figure 8 to Appendix C

These updates are highlighted in blue text for your convenience.
We hope our response and revision sincerely address all the reviewers’ concerns.

Best regards,

Authors.

---

### Meta-Review · Area_Chair_dP3A · 2023-12-05

**Metareview:**

- Claims and findings:

This submission proposed a way to inject spatial features into MLPs in the context of learning neural fields. Experimental results demonstrate that this approach enhances the performance of neural field representation and improves learning stability across a range of signals. In novel view synthesis, the proposed approach achieved state-of-the-art performance with the least number of parameters and fast training speed for dynamic scenes and the best performance under 1MB memory for static scenes. The approach also outperforms the best-performing video compression methods using neural fields by a large margin.

- Strengths:

Reviewers have pointed out that authors are tackling a very relevant problem, with wide interest to practitioners and that the paper is well written and easy to follow. In addition, reviewers have highlighted that the the argument about spectral bias and not learning high frequency components is verified empirically in Section 4.3. Reviewers have also pointed out that experiments are sound and cover a wide range of tasks (images, video, view synthesis, etc.). Results are strong with performance comparable or exceeding the state of the art.

- Weaknesses:

Most of the weaknesses pointed out by reviewers have been addressed in the rebuttal, including missing references, speed/inference memory required and clarity in the manuscript (eg. Fig 1 and 2).

- Missing in paper
N/A

**Justification For Why Not Higher Score:**

My reason for not having a higher score is that the impact of fitting neural fields is still unclear. It does require a non trivial amount of computation and it does not really pose a ML problem in terms of generalization to unseen coordinates. I believe the community would benefit from this paper but I wouldn't score it as an oral.

**Justification For Why Not Lower Score:**

The community will benefit from highlighting this work with a spotlight as it can benefit practitioners in a quite straightforward way.

---

### Decision · Program_Chairs · 2024-01-16

Accept (spotlight)